# Correlates of Poor Self-Assessed Health Status among Socially Disadvantaged Populations in Poland

**DOI:** 10.3390/ijerph17041372

**Published:** 2020-02-20

**Authors:** Joanna Jurewicz, Dorota Kaleta

**Affiliations:** Department of Hygiene and Epidemiology, Medical University of Lodz, 7/9 Żeligowskiego St, 90-752 Łódź, Poland; dkaleta@op.pl

**Keywords:** self-rated health, sociodemographic correlates, lifestyle factors, health conditions

## Abstract

Self-assessment of health is recommended as valuable source of information about subjective health status. The present study was performed to evaluate the correlates of self-rated health status among beneficiaries of social care in Poland. This assessment could be crucial for the implementation of targeted preventive measures among this valuable population. The study population consisted of 1710 beneficiaries of social care from the Piotrkowski District. The relationship between self-rated health status and its correlates (sociodemographic, lifestyle factors, and health conditions) was examined using logistic regression, with a poor health rating as the outcome. Overall, 11% of respondents declared poor self-assessed health status. Men more often rated health status as poor (15%) as compared to women (8.5%) (*p* < 0.001). The odds of a poor assessment of health increased with age, being unemployed or disabled/retired (OR = 2.34 95%CI (1.34–4.19) or OR = 9.07 95%CI (3.68–22.37), respectively), and additionally with poor life satisfaction (OR = 5.14 95% CI (1.94–13.64)). Regarding lifestyle characteristics, only binge drinking was associated with poor health status assessment (OR = 12.62 95%CI (3.71–42.87)). In addition, having any illness or health problems decreased health status (OR = 4.26 95%CI (1.36–13.31)). Socially-disadvantaged populations, especially men who poorly rated their health status, still constituted a large percentage of the population, which is an important public health problem. Increasing knowledge about the correlates of health status will allow greater prevention strategies to be developed for the population.

## 1. Introduction

The World Health Organization (WHO) defines health as not only the absence of disease, but also as a broader sense of complete physical, emotional, and social well-being at the individual, family, or community level (WHO 1948) [1]. Health is affected not only by risk factors and unhealthy behaviors, but also by economic and social conditions.

Socioeconomic status has been identified in numerous studies as an important risk factor in the occurrence of a disease [2,3]. People who are poor, powerless, and less educated have more health problems and shorter lifespans than those more educated and with higher income.

To improve health equality and provide more patient-oriented care, it is necessary to better recognize, understand, and address correlates and predictors of poor health.

Self-rated health (SRH) is considered to be a valuable source of data to examine the health problems in various populations [4,5]. Self-rated health status seems to reflect not only biological, but also psychological and social aspects of health, so it is a comprehensive perception of health [4]. SRH has been recognized as a reliable and valid health indicator that is based on a simple question in which the respondents are asked to rate their current general health status [5]. As the data about SRH are easily collected, this tool is frequently used in epidemiological studies assessing health conditions [4].

Studies based on self-rated health status performed in various populations from different countries have confirmed that SRH is an important determinant of mortality [6,7,8]. The participants who reported to be in poor health had higher odds of death, 2–7 times greater than those who perceived their health status as excellent or very good [5,9,10]. Some studies have found evidence of an association between self-rated health and morbidity [11], lifestyle [9], and socio-demographic factors [12]. Therefore, self-assessment of health may be important in the estimation of risk factors among people with low socioeconomic status [13]. The determinates of SRH, such as lifestyle factors or specific health conditions, are important in terms of specific targeted prevention. Knowledge about the correlates of SRH can help public health professionals prioritize health promotion, education, and disease prevention interventions. Additionally, such information is needed for developing appropriate public health policies and programs to improve the overall health of the population. Public health strategies to promote healthy lifestyles or disease prevention interventions can be performed to improve personal health.

There is a difference between the quality of healthcare received by individuals with high or low socio-economic status. Individuals with a low income or education are less likely to attend cancer screening than better-educated and wealthier individuals [14]. Socio-economically disadvantaged individuals have poorer health [15]; with higher rates of obesity and alcohol consumption [16], ischemic heart disease [17], type two diabetes, and other chronic health problems [18]; and greater chances of premature mortality [19].

The aims of this study were to analyze the association between correlates of self-rated health status among adult social care beneficiaries in Poland. Such predictors have not been studied among this special Polish population. Additionally, according to our knowledge, this is one of the first studies to examine the different health predictors (sociodemographic factors, lifestyle characteristics, and health conditions) in such a complex way. In the present study, the subjective measure of health status is the main dependent variable, while the sociodemographic characteristics, objective health status, lifestyle factors, and satisfaction with life are correlates or predictors.

The correlates of self-assessed poor health status will help to gain knowledge about health problems and will help social assistance beneficiaries to adjust the healthcare and health counseling to their needs. This could be crucial for the implementation of targeted preventive measures among this valuable population. The utility of a simple formula for self-rating health can be an important screening tool to quickly identify the person at risk among social assistance beneficiaries.

## 2. Materials and Methods

### 2.1. Study Population

This analysis is a part of data collected in the study entitled “Reducing Social Inequalities in Health” [20]. The details of the study were previously described [20,21,22]. Briefly, the study population consisted of social care beneficiary adults aged 18–59 years from Piotrkowski District. This District has a low index of development, especially social development, for which is ranked at 11th place in Poland. Among 11,867 social care beneficiaries in Piotrkowski District, 3636 people were in the age category 18–59 years and 50% of them agreed to participate in the study (*n* = 1817). The information about health status was available from 1710 participants.

The Medical University in Lodz Bioethical Committee Board approved the study protocol, and written informed consent was received from each study subject before their participation.

### 2.2. Collected Data and Measurements

Face-to-face interviews were conducted. The questionnaire included questions regarding sociodemographic characteristics, lifestyle factors, and health problems [20,21,22].

Respondents were asked to assess their health status based on the question “assess your current health status” and were offered answers including: “Fair”, “rather fair”, “neither fair nor poor”, “rather poor”, and “poor”. In the analysis, the categories “fair” and “rather fair” were treated as one—fair—whereas the categories “rather poor” and “poor” were assessed as poor. Additionally, the health problems declared by the study participants were analyzed as none, between 1 and 3, between 4 and 6, and more than 7.

### 2.3. Statistical Analysis

Descriptive statistics for respondents grouped by socio-demographic characteristics, along with the distributions of lifestyle factors, health conditions, and subjective health status assessment, were calculated. Mantel–Haenszel chi-square statistics were used to assess the correlates of self-rated health status. The relationship between poor self-rated health status and all of the examined variables (sociodemographic characteristics, lifestyle factors, health condition) were estimated before and after controlling for potential confounding factors. All of the variables significantly associated with poor self-rated health in univariate models (*p* ≤ 0.05) were included in the multivariate logistic regression analysis.

The following variables were treated as confounders: Gender, age, smoking, education, employment status, subjective assessment of income, living conditions, life satisfaction, current smoking, binge drinking, and selected health problems. The significance level of statistical inference was set at *p* < 0.05. The analysis was based on STATISTICA Windows XP version 10.0 (StataSoft Poland Inc., Tulsa, OK, USA).

## 3. Results

### 3.1. Study Population Characteristics

A total of 1710 beneficiaries of social care were included in the current study—1142 (66.8%) women and 568 (33.2%) men. Excluded from the analysis were participants with missing information about health status. In Table 1, the socio-demographic characteristics of the study population are presented. In summary, most of the participants had vocational (33.1%) and secondary (34.1%) education. The mean (±SD (standard deviation)) age was 41.1 years of age among men and 38.2 years among women (*p* < 0.001). The permanent occupational activity was declared by 30% of the study subjects, while 58.4% were unemployed. Of the study subjects, 84% were cohabitating with a partner and/or family, and 52.3% of the study participants declared that their income was sufficient only for basic needs. About 25% of respondents reported insufficient income to cover even basic needs (*p* < 0.001). Living conditions were assessed in 46% as fair and rather fair and in 45% as neither fair nor poor (*p* = 0.01).

### 3.2. Lifestyle Characteristics among Study Participants

Most of the respondents (76%) reported to not drink alcohol at all. The consumption of alcohol was more frequent among men than women (*p* < 0.001) (Table 2). The most common alcoholic drink was beer (68% males and 32% females; *p* < 0.001). The frequency of beer drinking was a few times a month for men (64.3%) and less than once a year for women (65%). Other alcoholic beverages (wine, spirits) were not as frequently used, and fewer participants declared drinking them. Additionally, binge drinking was more frequent among men compared to women (*p* < 0.001). Leisure-time physical activity was more popular in females compared to men (*p* < 0.001). Almost 37% of respondents reported current smoking, which was more frequent among men (*p* < 0.001). Only 3% of the study participants declared healthy dietary habits. Men and women did not differ significantly in reporting unhealthy dietary habits (*p* > 0.05). Additionally, satisfaction with daily life was reported as neither fair nor poor (neutral) by most of the study participants (50%).

### 3.3. Health Status among Study Participants

Almost 66% of respondents rated their health status as good and fairly good (Table 3), whereas 86% declared one of the stated health problems. Over half (54%) reported as many as 1–3 health problems, 27% 4–6 health problems, and 6% 7 or more health complications. Almost 12% of study participants reported that they had high blood pressure, most of whom were women (59%). Diabetes and heart attacks were declared by 2.5% and 1.3% of social care beneficiaries, respectively. There was no statistically significant difference between selected diseases (diabetes and heart attack) among men and women (*p* > 0.05).

### 3.4. Correlates of Self-Assessed Health Status

Poor health status was mostly declared by men (*p* < 0.001), whereas fair health status was more commonly reported among women (*p* < 0.001) (Table 4). Poor and neither fair nor poor ratings of health increased with age (*p* < 0.001) and decreased with education level (*p* < 0.001), whereas fair health status decreased with age (*p* < 0.001) and increased with the education level. Unemployed, disabled, or retired respondents assessed their health as poor or neither poor nor fair (*p* < 0.001 and *p* < 0.02, respectively). Subjective monthly income assessment was in line with employment status. Study subjects with poor health mostly declared that their income was only meeting basic needs (*p* < 0.001). Income assessed as sufficient to cover all living needs was mostly declared by participants with fair health status (*p* < 0.001). Living conditions and life satisfaction were subjectively rated as poor among study subjects with poor or neutral health status (*p* < 0.001), whereas fair assessed health participants were satisfied with life and living conditions (*p* < 0.001).

Participants currently smoking and binge drinking assessed their health as poorer than did other participants (*p* < 0.02 and *p* < 0.002, respectively). Additionally, among participants with poor and neither fair nor poor health status, the number of health problems increased. Reported heart attacks, diabetes, and high blood pressure were also statistically significant among participants who assessed their health status as poor (Table 4).

Self-rated health status was not significantly associated with dietary quality scores, alcohol consumption, cohabitation with a partner and/or family, and physical activity (Table 4). All examined correlates concerning self-rated health status are presented in Table 4.

The results of the logistic regression analyses for poor self-assessed health status with socio-demographic, lifestyle, and health correlates are presented in Table 5. Most of the examined sociodemographic characteristics were significantly associated with poorly self-assessed health. Male beneficiaries with lower education, temporary jobs, who were unemployed, disabled, or retired, with a monthly income that was sufficient to cover living and basic needs, with rather poor, neither fair nor poor, or poor living conditions, and who were not satisfied with their life, had higher odds of self-assessment of their health status as poor. In addition, current smoking, binge drinking (at least once a week), reported health problems, and examined diseases (heart attack, high blood pressure, diabetes) were statistically significantly related to poor assessment of health status only in the univariate model.

When the model was fully adjusted, including all statistically significant characteristics, respondents in the age categories of 30–39, 40–49, and 50–59 years of age had higher odds of poor assessment of health status than respondents in the age category of <30 years of age (5.82 95% CI (1.25–27.16), 10.29 95% CI (2.21–47.89), 10.68 95% CI (2.24–50.75), respectively) (Table 5). Being a male was also associated with almost double the odds for poor assessment of health status. The disabled or retired participants had 9 times higher odds of poor self-assessment of health status than subjects with a permanent job (OR = 9.07 95% CI (3.68–22.37)). Unemployed subjects also had double the odds of poor assessment of health status compared to subjects with permanent jobs (2.34 95% CI (1.34–4.19)).

Binge drinking at least once a week was associated with higher odds of poor health status assessment (12.62 95% CI (3.71–42.87)) than not drinking at all. Study participants who reported life satisfaction as neutral, slightly dissatisfied, and dissatisfied/extremely dissatisfied had higher odds of for poorly assessing health status (2.38 95% CI (1.35–4.20), 5.06 95% CI (2.41–10.63), 5.14 95% CI (1.94–13.64), respectively).

The declaration of more than one health complaint was associated with poor self-assessment of health status. Participants reporting 1–3 health complaints, 4–6, and 7 or more had almost 5, 18, and 40 times higher odds of poor self-assessment of health status than not reporting any health condition. Additionally, subjects with selected diseases, such as heart attack and high blood pressure, had higher odds for poor assessment of health status (OR = 4.25 95% CI (1.36–13.31) and OR = 1.92 95% CI (1.18–3.15), respectively) compared to subjects without those diseases (Table 5).

## 4. Discussion

Our analysis describes differences in self-rated health status among a socially disadvantaged population. According to our knowledge, this is the first study to examine the SRH among this special population of beneficiaries of social care in Poland. Additionally, this study assessed many different correlates of health status (sociodemographic and lifestyle factors, and health conditions) in a complex way.

In this study, we found that sociodemographic characteristics, as well as lifestyle and health factors, are associated with poor self-rated health status.

Self-rated health status was lower in men than women, which is similar to the results of a study performed in Estonia [23]. Estonian men have poorer self-rated health than women [23]. Previously published studies were not consistent, with some performed in the United Kingdom showing that women tend to rate their health lower than men [24,25,26], and others performed in countries of the former Soviet Union reporting better self-rated health for women [27], or else no gender differences [28]. Additionally, in all age categories, the odds for rating health status as poor were higher compared to the group <30 years of age. This is in line with different studies performed in Denmark, the United Kingdom, and Iceland, where self-assessed health decreases with age [29,30,31]. In general, older groups may have had more disabilities and health conditions that impact on the poor rating of health status.

In this study, poor self-rated health was significantly higher among the unemployed, disabled, or retired participants than among those who were employed. There is a link between unemployment and poor self-rated health status in many studies [32,33]. Unemployment may be a physiological or psychological risk factor for poor health. Additionally, direct health consequences of unemployment may exist, such as symptoms of anxiety and depression, or having chronic somatic conditions [32]. Nevertheless, this is one of the risk factors for more poorly-rated health status. Moreover, poor self-rated health is an important determinant of disability/retirement [34].

Binge drinking at least once a week and poor life satisfaction are correlates of poor health status assessment. Other lifestyle factors, such as smoking cigarettes, drinking alcohol, physical activity, and dietary habits were not associated with poorer self-rated health. In most of the previously published papers, these lifestyle factors reduced the reporting of health status as excellent or very good [35]. This may be associated with the study population, which is different than in other studies. In the present study, the participants were social care beneficiaries from one district. Alternatively, in the study performed by Shields and Shooshtari [35], the study population consisted of Canadian residents in all provinces and territories.

The association between alcohol consumption and self-rated health varies between countries. In Scandinavian countries and in the United Kingdom, rating health status as good was less frequent among moderate drinkers and more frequent in moderate drinkers than among nondrinkers [36,37], whereas in Mediterranean countries, moderate and even excessive drinkers have a lower frequency of poorly rating health status than never-drinkers [38]. Our findings indicate that increased levels of binge drinking may place some drinkers at a greater risk of poorer assessed health. These findings are consistent with those from similar studies in different populations [39,40,41].

We found that poor life satisfaction is associated with a poor self-rating of health, indicating that self-rated health may impact life satisfaction (LS), which is in line with previously published studies. These indicators are closely linked and considered essential components of quality of life [42]. LS, as well as SRH, are seen to be important components of broader strategies oriented to improve health and overall quality of life for people [43].

Most of the previously published studies suggested that physical activity is an important predictor of good self-rated health. An international study reported that physical activity is positively associated with self-rated health [44]. In the present study, we did not find such an association because of the special population examined mostly focused on current, basic needs.

Additionally, no relationship was found between cigarette smoking and SRH, which is different than in the literature which suggests such a relationship [45]. Only in the univariate model was smoking associated with a poor rating of health, which suggests that other variables are more important in the poor assessment of health among study population, e.g., unemployment, age, or health condition. Additionally, no relationship was found between dietary habits and SRH. We hypothesized that cross-cultural differences might account for the differences in these results.

We found that having any illness or health problem decreased health status. Poor self-rated health status was determined by more than one health complaint and selected diseases (cardiovascular diseases). This is in line with previously published studies that suggest that poor assessment may be related to some physical problems [46,47,48] or chronic health conditions [49,50]. Social care beneficiaries more often poorly assessed their health status than residents of socioeconomically-advantaged populations.

Our study is not without limitations. First, the study used a cross-sectional design, which provides measures at only one period in time, and may not adequately represent measures of self-rated health and other measures that can change over time. Second, this study was limited to the Piotrkowski District, which may limit the generalizability of the findings to the general population of Poland.

The major strengths of this study are that the study was performed among a special population of socially-disadvantaged adults; additionally, participation rate was about 50% and face-to-face interviews were completed.

In the study, we used self-rated health status assessment. Self-perceived health is a reliable and valid measure, which is an important issue for population health screening [35]. Self-rated health is recognized as a good and important predictor of chronic health problems and/or psychological well-being [51] in different populations [51,52].

## 5. Conclusions

Socially-disadvantaged populations, especially men who poorly rated their health status, still constituted a large percentage of the population. The results indicate that there are some correlates or predictors of self-rated health status among beneficiaries of government welfare assistance.

These correlates include sociodemographic characteristics (age, gender, employment status), and lifestyle (alcohol consumption, poor life satisfaction) and health factors (illnesses, health problems, chronic diseases). The understanding of these factors is crucial to adjust health promotion, education, and care to special needs. The correlates of SRH are important in terms of specific targeted prevention. This is especially valid for special populations, such as beneficiaries of government welfare assistance, which are ignored or overlooked in most of the public health programs.

The results may have implications for public health policy, as many of the predictors influencing SRH can be reduced or changed by specific interventions. The knowledge about the correlates of SRH can help public health professionals prioritize health promotion, education, disease prevention interventions, but also social care, to improve the overall health of the population. This suggests the need for actions aimed at the prevention and diagnosis of chronic diseases (e.g., cardiovascular diseases), promotion of a healthy lifestyle (e.g., reducing alcohol consumption), and reduction of inequalities (e.g., unemployment).

Our findings suggest that SRH may be a simple, low-cost measure used in epidemiological and public health studies for health status monitoring in different populations. This simple model can be an important screening tool to identify the person at risk. Additionally, information about health status based on SRH can be used by national authorities to develop appropriate public health policies and programs, to allocate the resources effectively, and to identify the areas or populations that require special attention. When starting to promote healthy lifestyles and direct efforts towards the prevention of chronic diseases, we should take into account the differences between the general population and socially-disadvantaged individuals.

## Figures and Tables

**Table 1 ijerph-17-01372-t001:** Characteristics of the study participants.

Variable	Total *N* = 1710	Men *N* = 568	Women *N* = 1142	*p*-Value
**Age (years)**				
Mean ± SD	39.2 ± 7.7	41.1 ± 8.1	38.2 ± 7.2	<0.001
<30	194 (11.3%)	47 (27.7%)	147 (72.3%)	<0.001
30–39	725 (42.4%)	201 (36.5%)	524 (63.5%)	
40–49	578 (33.8%)	211 (37.1%)	367 (32.1%)	
50–59	213 (12.5%)	109 (51.2%)	104 (48.8%)	
**Education**				
Primary	468 (27.4%)	204 (43.6%)	264 (56.4%)	<0.001
Vocational	566 (33.1%)	228 (40.3%)	338 (59.7%)	
Secondary	583 (34.1%)	128 (22%)	455 (78%)	
High	93 (5.4%)	8 (8.6%)	109 (91.4%)	
**Employment status**				
Permanent job	507 (29.6%)	215 (42.4%)	292 (57.6%)	<0.001
Temporary job	149 (8.7%)	70 (47%)	79 (53%)	
Disabled or retired	55 (3.2%)	28 (50.9%)	27 (49.1%)	
Unemployed	999 (58.4%)	255 (25.5%)	744 (74.5%)	
**Subjective assessment of monthly income**				
Sufficient to cover all living needs plus may save a certain amount	208 (12.2%)	57 (27.4%)	151 (72.6%)	<0.001
Sufficient to cover all living needs	894 (52.3%)	275 (30.8%)	619 (69.2%)	
Sufficient to cover basic needs only	433 (25.3%)	183 (42.3%)	250 (57.7%)	
Difficult to say	175 (10.2%)	53 (30.3%)	122 (69.7%)	
**Subjective assessment of living conditions**				
Fair/rather fair	787 (46.0%)	231 (29.4%)	556 (70.6%)	0.01
Neither fair nor poor	774 (45.3%)	284 (36.7%)	490 (63.3%)	
Rather poor	85 (5.0%)	28 (32.9%)	57 (67.1%)	
Very poor	30 (1.7%)	14 (46.7%)	16 (53.3%)	
Difficult to say	34 (2.0%)	11 (32.4%)	23 (37.6%)	
**Cohabitation with partner and/or family**				
Yes	1444 (84.4%)	479 (33.2%)	965 (66.8%)	>0.05
No (living alone)	266 (15.6%)	89 (33.5%)	177 (66.5%)	

SD: standard deviation.

**Table 2 ijerph-17-01372-t002:** Lifestyle factors among study participants.

Variable	Total	Men	Women	*p*-Value
**Alcohol consumption**				
**Spirits**				
Every day	2 (0.1%)	2 (100%)	0 (0.0%)	<0.001
Few times per week	18 (1.0%)	13 (72.2%)	5 (27.8%)	
Few times per month	87 (5.1%)	71 (81.6%)	16 (18.4%)	
Less than once per year	302 (17.7%)	144 (47.7%)	158 (52.3%)	
Never	1301 (76.1%)	338 (26.0%)	963 (74.0%)	
**Wine**				
Every day	1 (0.1%)	1 (100.0%)	0 (0.0%)	<0.001
Few times per week	8 (0.5%)	6 (75.0%)	2 (25.0%)	
Few times per month	50 (2.9%)	16 (32.0%)	34 (68.0%)	
Less than once per year	290 (17.0%)	46 (15.9%)	244 (84.1%)	
Never	1361 (79.6%)	499 (36.7%)	862 (63.3%)	
**Beer**				
Every day	5 (0.3%)	5 (100.0%)	0 (0.0%)	<0.001
Few times per week	68 (4.0%)	55 (80.9%)	13 (19.1%)	
Few times per month	263 (15.4%)	169 (64.3%)	94 (35.7%)	
Less than once per year	339 (19.8%)	118 (34.8%)	221 (65.2%)	
Never	1035 (60.5%)	221 (21.4%)	814 (78.6%)	
**Others**				
Every day	0 (0.0%)	0 (0.0%)	0 (0.0%)	>0.05
Few times per week	4 (0.2%)	4 (100.0%)	0 (0.0%)	
Few times per month	5 (0.3%)	3 (60.0%)	2 (40.0%)	
Less than once per year	34 (2.0%)	11 (32.4%)	23 (67.6%)	
Never	1667 (97.5%)	550 (33.0%)	1117 (67.0%)	
**Alcohol consumption (if yes in each frequency category)**				
**Spirits**				
Yes	107 (6.3%)	86 (80.4%)	21 (19.6%)	<0.001
No	1603 (93.7%)	482 (30.1%)	1121 (69.9%)	
**Wine**				
Yes	59 (3.5%)	23 (39.0%)	36 (61.0%)	>0.05
No	1651 (96.5%)	545 (33.0%)	1106 (67.0%)	
**Beer**				
Yes	336 (19.6%)	229 (68.1%)	107 (31.9%)	<0.001
No	1374 (80.4%)	339 (24.7%)	1035 (75.3%)	
**Alcohol consumption (if yes in each frequency category of different alcohol type)**				
Yes	403 (23.6%)	261 (64.8%)	142 (35.2%)	<0.001
No	1307 (76.4%)	307 (23.5%)	1000 (76.5%)	
**Binge drinking**				
Never	1223 (71.5%)	239 (19.5%)	984 (80.5%)	<0.001
Few times per year	299 (17.5%)	212 (70.9%)	87 (29.1%)	<0.001
Once per month	39 (2.3%)	30 (76.9%)	9 (23.1%)	
Once per week	18 (1.1%)	18 (100.0%)	0 (0.0%)	
Few times per week	6 (0.3%)	5 (83.3%)	1 (16.7%)	
Don’t know	125 (7.3%)	64 (51.2%)	61 (48.8%)	
**Recreational physical activity**				
Yes	750 (43.9%)	216 (28.8%)	534 (71.2%)	<0.001
No	960 (56.1%)	352 (36.7%)	608 (63.3%)	
**Ever smoking**				
Yes	899 (52.6%)	395 (47.0%)	504 (53.0%)	<0.001
No	811 (47.4%)	173 (36.5%)	638 (63.5%)	
**Current smoking**				
Yes	527 (30.8%)	276 (23.1%)	251 (76.9%)	<0.001
Yes occasionally	110 (6.4%)	24 (28.7%)	86 (71.3%)	
No	1073 (62.8%)	268 (33.9%)	805 (66.1%)	
**Diet (Dietary Quality Score)**				
Healthy dietary habits	52 (3.0%)	12 (23.1%)	40 (76.9%)	>0.05
Average dietary habits	108 (6.3%)	31 (28.7%)	77 (71.3%)	
Unhealthy dietary habits	1550 (90.7%)	525 (33.9%)	1025 (66.1%)	
**Subjective assessment of life satisfaction**				
Extremely satisfied/Satisfied	702 (41.1%)	214 (30.5%)	488 (69.5%)	0.02
Neutral	855 (50.0%)	291 (34.0%)	564 (66.0%)	
Slightly dissatisfied	106 (6.2%)	39 (36.8%)	67 (63.2%)	
Dissatisfied/Extremely dissatisfied	47 (2.7%)	24 (51.1%)	23 (48.9%)	

**Table 3 ijerph-17-01372-t003:** Health status of the study participants.

**Variable**	Total	Men	Women	*p*-Value
**Subjective health state**				
Fair/rather fair	1121 (65.5%)	339 (30.2%)	782 (69.8%)	<0.001
Neither fair nor poor	407 (23.8%)	144 (35.4%)	263 (64.6%)	
Rather poor/poor	182 (10.6%)	85 (46.7%)	97 (53.3%)	
**Health problems (if yes at least in one health problem)**				
Yes	1445 (86.2%)	457 (31.6%)	988 (68.4%)	<0.001
No	231 (13.8%)	104 (45.0%)	127 (55.0%)	
**Number of health problems**				
None	231 (13.8%)	104 (45.0%)	127 (55.0%)	<0.001
1–3	900 (53.7%)	309 (34.3%)	591 (65.7%)	
4–6	448 (26.7%)	120 (26.8%)	328 (73.2%)	
>7	97 (5.8%)	28 (28.9%)	69 (71.1%)	
**Heart attack**				
Yes	22 (1.3%)	11 (50.0%)	11 (50.0%)	>0.05
No	1688 (98.7%)	557 (33.0%)	1131 (67.0%)	
**High blood pressure**				
Yes	197 (11.5%)	80 (40.6%)	117 (59.4%)	<0.02
No	1513 (88.5%)	488 (32.2%)	1025 (67.8%)	
**Diabetes**				
Yes	42 (2.5%)	16 (38.1%)	26 (61.9%)	>0.05
No	1668 (97.5%)	552 (33.1%)	1116 (66.9%)	

**Table 4 ijerph-17-01372-t004:** Subjective health state and the characteristics of the study population.

	Subjective Health State					
	Fair/Rather Fair	*p*-Value	Neither Fair nor Poor	*p*-Value	Rather Poor/Poor	*p*-Value
**Gender**						
Female	782 (68.5%)	<0.001	263 (23.0%)	>0.05	97 (8.5%)	<0.001
Male	339 (59.7%)		144 (25.4%)		85 (15.0%)	
**Age (years of age)**						
<30	160 (82.5%)	<0.001	31 (16.0%)	<0.001	3 (1.6%)	<0.001
30–39	557 (76.8%)		125 (17.2%)		43 (5.9%)	
40–49	321 (55.5%)		176 (30.5%)		81 (14.0%)	
50–59	83 (39.0%)		75 (35.2%)		55 (25.8%)	
**Education**						
Primary	265 (56.6%)	<0.001	133 (28.4%)	<0.001	70 (15.0%)	<0.001
Vocational	354 (62.5%)		148 (26.2%)		64 (11.3%)	
Secondary	423 (72.6%)		115 (19.7%)		45 (7.7%)	
High	79 (84.9%)		11 (11.8%)		3 (3.2%)	
**Employment status**						
Permanent job	393 (77.5%)	<0.001	96 (18.9%)	<0.02	18 (3.6%)	<0.001
Temporary job	95 (63.8%)		42 (28.2%)		12 (18.0%)	
Disabled or retired	23 (41.8%)		12 (21.8%)		20 (36.4%)	
Unemployed	610 (61.1%)		257 (25.7%)		132 (13.2%)	
**Subjective assessment of monthly income**						
Sufficient to cover all living needs plus may save a certain amount	183 (88.0%)	<0.001	20 (9.6%)	<0.001	5 (2.4%)	<0.001
Sufficient to cover all living needs	604 (67.6%)		220 (24.6%)		70 (7.8%)	
Sufficient to cover basic needs only	202 (46.6%)		137 (31.6%)		94 (21.7%)	
Difficult to say	132 (75.4%)		30 (17.1%)		13 (7.4%)	
**Subjective assessment of living conditions**						
Fair/rather fair	602 (76.5%)	<0.001	138 (17.5%)	<0.001	47 (6.0%)	<0.001
Neither fair nor poor	454 (58.7%)		226 (29.2%)		94 (12.1%)	
Rather poor	33 (38.8%)		29 (34.1%)		23 (27.1%)	
Very poor	12 (40.0%)		8 (26.7%)		10 (33.3%)	
Difficult to say	20 (58.8%)		6 (17.7%)		8 (23.5%)	
**Subjective assessment of life satisfaction**						
Extremely satisfied/Satisfied	579 (82.5%)	<0.001	100 (14.2%)	<0.001	23 (3.3%)	<0.001
Neutral	498 (58.2%)		251 (29.4%)		106 (12.4%)	
Slightly dissatisfied	36 (34.0%)		36 (34.0%)		34 (32.0%)	
Dissatisfied/Extremely dissatisfied	8 (17.0%)		20 (42.6%)		19 (40.4%)	
**Cohabitation with partner and or family**						
Yes	942 (65.2%)	>0.05	352 (24.4%)	>0.05	150 (10.4%)	>0.05
No	179 (67.3%)		55 (20.7%)		32 (12.0%)	
**Current smoking**						
Yes	318 (60.3%)	<0.02	138 (26.2%)	>0.05	71 (13.5%)	<0.02
Yes, occasionally	74 (67.3%)		22 (20.0%)		14 (12.7%)	
No	729 (67.9%)		247 (23.1%)		97 (9.0%)	
**Physical activity**						
Yes	498 (66.4%)	>0.05	176 (23.5%)	>0.05	76 (10.1%)	>0.05
No	623 (64.9%)		231 (24.1%)		106 (11.0%)	
**Alcohol consumption**						
Yes	262 (65.0%)	>0.05	94 (23.3%)	>0.05	47 (11.7%)	>0.05
No	859 (65.7%)		313 (24.0%)		135 (10.3%)	
**Binge drinking**						
Never	798 (65.3%)	>0.05	298 (24.4%)	>0.05	127 (10.4%)	<0.002
Few times per year	191 (63.9%)		80 (26.8%)		28 (9.4%)	
Once per month	29 (74.4%)		6 (15.4%)		4 (10.3%)	
Once per week	9 (50.0%)		3 (16.7%)		6 (33.3%)	
Don’t know	91 (72.8%)		20 (16.0%)		14 (11.2%)	
**Number of health problems**						
None	212 (91.8%)	<0.001	16 (6.9%)	<0.001	3 (1.3%)	<0.001
1–3	690 (76.7%)		166 (18.4%)		44 (4.9%)	
4–6	184 (41.1%)		174 (38.8%)		90 (20.1%)	
>7	20 (20.6%)		40 (41.2%)		37 (38.1%)	
**Heart attack**						
Yes	3 (13.6%)	<0.001	9 (40.9%)	<0.001	10 (45.5%)	<0.001
No	1118 (66.2%)		398 (23.6%)		172 (10.2%)	
**High blood pressure**						
Yes	63 (32.0%)	<0.001	83 (41.1%)	<0.001	51 (25.9%)	<0.001
No	1058 (69.9%)		324 (21.4%)		131 (8.7%)	
**Diabetes**						
Yes	10 (23.8%)	<0.001	18 (42.9%)	<0.001	14 (23.3%)	<0.001
No	1111 (66.6%)		389 (23.3%)		168 (10.1%)	
**Diet (Dietary Quality Score)**						
Healthy dietary habits	36 (69.2%)	>0.05	13 (25.0%)	>0.05	3 (5.8%)	>0.05
Average dietary habits	80 (74.1%)		18 (16.7%)		10 (9.3%)	
Unhealthy dietary habits	1005 (64.8%)		376 (24.3%)		169 (10.9%)	

**Table 5 ijerph-17-01372-t005:** Correlates of poor self-rated health status.

	Total	Subjective Health Status	Univariate Logistic Regression	Multivariate Logistic Regression
		Rather Poor/Poor	OR 95% CI	OR 95% CI
**Gender**				
Female	1142 (66.8%)	97 (8.5%)	1.0 Ref.	1.0 Ref.
Male	568 (33.2%)	85 (15.0%)	1.90 *** (1.39–2.59)	1.79 * (1.12–2.89)
**Age (years of age)**				
<30	194 (11.3%)	3 (1.6%)	1.0 Ref.	1.0 Ref.
30–39	725 (42.4%)	43 (5.9%)	4.01 * (1.23–13.09)	5.82 * (1.25–27.16)
40–49	578 (33.8%)	81 (14.0%)	10.38 *** (3.24–33.26)	10.29 ** (2.21–47.89)
50–59	213 (12.5%)	55 (25.8%)	22.16 ***(6.79–72.32)	10.68 ** (2.24–50.75)
**Education**				
Primary	468 (27.4%)	70 (15.0%)	5.28 ** (1.62–17.15)	2.34 (0.49–11.28)
Vocational	566 (33.1%)	64 (11.3%)	3.82 * (1.18–12.44)	2.16 (0.45–10.30)
Secondary	583 (34.1%)	45 (7.7%)	2.51 (0.76–8.25)	2.37 (0.50–11.32)
High	93 (5.4%)	3 (3.2%)	1.0 Ref.	1.0 Ref.
**Employment status**				
Permanent job	507 (29.6%)	18 (3.6%)	1.0 Ref.	1.0 Ref.
Temporary job	149 (8.7%)	12 (18.0%)	2.38 * (1.12–5.06)	1.18 (0.49–2.85)
Disabled or retired	55 (3.2%)	20 (36.4%)	15.52 *** (7.53–32.02)	9.07 *** (3.68–22.37)
Unemployed	999 (58.4%)	132 (13.2%)	4.14 *** (2.50–6.6.85)	2.34 ** (1.34–4.19)
**Subjective assessment of monthly income**				
Sufficient to cover all living needs plus may save a certain amount	208 (12.2%)	5 (2.4%)	1.0 Ref.	1.0 Ref.
Sufficient to cover all living needs	894 (52.3%)	70 (7.8%)	3.45 ** (1.37–8.66)	1.50 (0.55–4.10)
Sufficient to cover basic needs only	433 (25.3%)	94 (21.7%)	11.26 *** (4.50–28.16)	2.00 (0.70–5.69)
Difficult to say	175 (10.2%)	13 (7.4%)	3.26 * (1.14–9.33)	1.73 (0.54–5.51)
**Subjective assessment of living conditions**				
Fair/rather fair	787 (46.0%)	47 (6.0%)	1.0 Ref.	1.0 Ref.
Neither fair nor poor	774 (45.3%)	94 (12.1%)	2.18 *** (1.51–3.14)	0.86 (0.54–1.38)
Rather poor	85 (5.0%)	23 (27.1%)	5.84 *** (3.33–10.25)	1.44 (0.66–3.14)
Very poor	30 (1.7%)	10 (33.3%)	7.87 *** (3.49–17.78)	1.05 (0.33–3.29)
Difficult to say	34 (2.0%)	8 (23.5%)	4.48 *** (2.08–11.28)	1.13 (0.89–8.99)
**Subjective assessment of life satisfaction**				
Extremely satisfied/Satisfied	702 (41.1%)	23 (3.3%)	1.0 Ref.	1.0 Ref.
Neutral	855 (50.0%)	106 (12.4%)	4.18 *** (2.63–6.64)	2.38 ** (1.35–4.20)
Slightly dissatisfied	106 (6.2%)	34 (32.0%)	13.94 *** (7.78–24.97)	5.06 *** (2.41–10.63)
Dissatisfied/Extremely dissatisfied	47 (2.7%)	19 (40.4%)	20.03 *** (9.79–41.00)	5.14 *** (1.94–13.64)
**Cohabitation with partner and/or family**				
Yes	1444 (84.4%)	150 (10.4%)	1.0 Ref.	
No	266 (15.6%)	32 (12.0%)	1.18 (0.79–1.77)	
**Current smoking**				
Yes	527 (30.8%)	71 (13.5%)	1.57 ** (1.13–2.17)	1.08 (0.69-1.69)
Yes, occasionally	110 (6.4%)	14 (12.7%)	1.47 (0.81–2.67)	1.55 (0.73-3.28)
No	1073 (62.8%)	97 (9.0%)	1.0 Ref.	1.0 Ref.
**Physical activity**				
Yes	750 (43.9%)	76 (10.1%)	1.0 Ref.	
No	960 (56.1%)	106 (11.0%)	1.11 (0.81–1.51)	
**Alcohol consumption**				
Yes	403 (23.6%)	47 (11.7%)	1.15 (0.81–1.63)	
No	1307 (76.4%)	135 (10.3%)	1.0 Ref.	
**Binge drinking**				
Never	1223 (71.5%)	127 (10.4%)	1.0 Ref.	1.0 Ref.
Few times per year	299 (17.5%)	28 (9.4%)	0.96 (0.62–1.48)	0.81 (0.46-1.42)
Once per month	39 (2.3%)	4 (10.3%)	1.06 (0.37–3.05)	0.28 (0.06-1.43)
Once per week	24 (1.4%)	9 (37.5%)	5.58 *** (2.39–13.05)	12.62 ***(3.71-42.87)
Difficult to say	125 (7.3%)	14 (11.2%)	1.17 (0.65–2.11)	0.94 (0.41-2.19)
**Number of health problems**				
None	231 (13.8%)	3 (1.3%)	1.0 Ref.	1.0 Ref.
1–3	900 (53.7%)	44 (4.9%)	3.91 * (1.20–12.70)	4.64 * (1.22-17.68)
4–6	448 (26.7%)	90 (20.1%)	19.11 *** (5.97–61.15)	17.69 *** (4.61–67.88)
>7	97 (5.8%)	37 (38.1%)	46.87 *** (13.91–157.84)	39.63 *** (9.52–165.09)
**Heart attack**				
Yes	22 (1.3%)	10 (45.5%)	7.34 *** (3.13–17.26)	4.26 * (1.36–13.31)
No	1688 (98.7%)	172 (10.2%)	1.0 Ref.	1.0 Ref.
**High blood pressure**				
Yes	197 (11.5%)	51 (25.9%)	3.69 *** (2.56–5.31)	1.92 ** (1.18–3.15)
No	1513 (88.5%)	131 (8.7%)	1.0 Ref.	1.0 Ref.
**Diabetes**				
Yes	42 (2.5%)	14 (23.3%)	4.46*** (2.30–8.65)	1.49 (0.61–3.66)
No	1668 (97.5%)	168 (10.1%)	1.0 Ref.	1.0 Ref.
**Diet (Dietary Quality Score)**				
Healthy dietary habits	52 (3.0%)	3 (5.8%)	1.0 Ref.	
Average dietary habits	108 (6.3%)	10 (9.3%)	1.67 (0.44–6.34)	
Unhealthy dietary habits	1550 (90.7%)	169 (10.9%)	2.00 (0.62–6.49)	

* *p* < 0.05; ** *p* < 0.01; *** *p* < 0.001; Fully-adjusted model, including all statistically significant characteristics. Ref - reference; CI - confidence interval.

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
