# Peer review of "This manuscript is a resubmission of an earlier submission. The following is a list of the peer review reports and author responses from that submission."

_ijerph, 2020, doi:10.3390/ijerph17041372_

Round 1

Reviewer 1 Report

The authors have presented a manuscript titled "Correlates of poor self-assessed health status among socially-disadvantaged populations in Poland". There are several concerns that need to be addressed in order to give the manuscript publication consideration.

There are no line numbers in the manuscript making it difficult to direct the authors to the exact site for revision.  

Abstract: Correct the sentence "The present study was carried out to evaluate the corelatecorrelates of self-rated health status among beneficiaries of social care in Poland"..... Use the terms 'performed' and 'correlates' respectively. Introduction: See highlighted texts in the attached manuscript for revision. The authors have not clearly defined the meaning of self-reported health. The concept is very vague, and leaves the reader to ponder the relevance of the current study. How does SRH specifically benefit the community in terms of disease prevention and health promotion? The text of the introduction is vague in this respect. Methods: What kind of interview did the study participants have?... Survey, telephone, online, face-to-face, etc? What was the response rate? How did the authors define health in terms of the classifications of 'poor', 'fair', 'rather fair', 'rather poor'? How were the participants able to understand their health status if no explanations of poor, fair, and good were stated? What is the source of the questionnaire, and what are the validity and reliability indices? Discussion: The authors failed to explain how the outcome of the study specifically benefit community/ public health. 

Author Response

Reviewer I

The authors have presented a manuscript titled "Correlates of poor self-assessed health status among socially-disadvantaged populations in Poland". There are several concerns that need to be addressed in order to give the manuscript publication consideration.

There are no line numbers in the manuscript making it difficult to direct the authors to the exact site for revision.  

Answer: The line numbers have been added.

Abstract: Correct the sentence "The present study was carried out to evaluate the corelatecorrelates of self-rated health status among beneficiaries of social care in Poland"..... Use the terms 'performed' and 'correlates' respectively.

Answer: The sentence has been corrected. The terms 'performed' and 'correlates' have been used.

Introduction: See highlighted texts in the attached manuscript for revision. The authors have not clearly defined the meaning of self-reported health. The concept is very vague, and leaves the reader to ponder the relevance of the current study. How does SRH specifically benefit the community in terms of disease prevention and health promotion? The text of the introduction is vague in this respect.

Answer: Thank you for highlighted remarks, I have changed it.

The information about the meaning of self-reported health has been added. Self-rated health (SRH) is considered to be valuable source of data to examine the health problems in various populations. SRH has been recognized as a reliable and valid health indicator that is based on a simple question in which the respondents are asked to rate their current general health status. As the data about SRH is easily collected this tool is frequently used in epidemiological studies assessing health conditions.

The information about how SRH specifically benefit the community in terms of disease prevention and health promotion have been explained. The determinates of SRH such as lifestyle factors or specific health conditions are important in terms of specific targeted prevention. The knowledge about the correlates of SHR can help public health professionals priorities health promotion, education and disease prevention interventions. Additionally such information is needed for developing appropriate public health policies and programmes to improve the overall health of the population. Public health strategies to promote healthy lifestyles or disease prevention interventions can be performed to improve personal health.

The text of the Introduction has been changed according to the Reviewer suggestions.

Methods: What kind of interview did the study participants have?... Survey, telephone, online, face-to-face, etc? What was the response rate? How did the authors define health in terms of the classifications of 'poor', 'fair', 'rather fair', 'rather poor'? How were the participants able to understand their health status if no explanations of poor, fair, and good were stated? What is the source of the questionnaire, and what are the validity and reliability indices?

Answer: The study population consists of social care beneficiaries adults aged 18–59 from Piotrkowski District. This District has a low index of development, especially social development and is in 11th place in Poland. Among 11,867 social care beneficiaries in Piotrkowski District 3636 people were in the age category 18–59 and 50% of them agreed to participate in the study (n=1817). The information about health status was available from 1710 participants. Face to face interviews were conducted with response rate 50%. All the requested information has been included.

The information about definition of health in terms of classification - Self-assessed general health (SAH) is one of the most frequently employed measure of health in epidemiological researches (Au and Johnston 2014, Idler 2000, 1990, 1997, Mackenbach 1994, DeSalvo 2006, Theme Filha 2008, Szwarcwald 2000). SRH has been recognized as a reliable and valid health indicator and is based on a simple question such as “In general how would you rate your health” (Au and Johnston 2014, Idler 2000, 1990, 1997, Mackenbach 1994, DeSalvo 2006, Theme Filha 2008, Szwarcwald 2000) and there is no explanation of each category. As the answer can be influenced by pain, depression, limited mobility or something else that is why it ts important to know the predictors of self-rated health to have the knowledge what impact on the answer. Respondents were asked to assess their health status based on the question ‘assess your current health status’ and were offered the available answers including: “fair”, “rather fair”, “neither fair nor poor”, “rather poor”, “poor”. Global self-ratings of health play an increasing role in the estimation of risk factors.

Reference:

Au N, Johnston DW. Self-assessed health: what does it mean and what does it hide? Soc Sci Med. 2014 Nov;121:21-8. doi: 10.1016/j.socscimed.2014.10.007. Epub 2014 Oct 5. Idler, E.L.; Kasl, S.V.; Lemke, J.H. Self-evaluated health and mortality among the elderly in New Haven, Connecticut, and Iowa and Washington counties, Iowa, 1982–1986. Am J Epidemiol. 1990, 131; 91-103. Mackenbach, J.P.; Van Den Bos, J.; Joung, I.M.A.; van de Mheen, H.; Stronks, K. The determinants of excellent health: Different from the determinants of ill-health? Int J Epidemiol 1994; 23(6), 1273-1281. Idler, E.L.; Benyamini, Y. Self-rated health and mortality: a review of twenty-seven community studies. J Health Soc Behav. 1997, 38; 21-37. DeSalvo, K.B.; Bloser, N.; Reynolds, K.; He, J.; Muntner, P. Mortality prediction with a single general self-rated health question. A meta-analysis. J Gen Intern Med. 2006, 21, 267-275. Theme Filha, M.M.; Szwarcwald, C.L.; Souza Junior, P.R. Measurements of reported morbidity and interrelationships with health dimensions. Rev Saude Publica. 2008, 42, 73-81. Szwarcwald, C.L.; Souza-Júnior, P.R.; Esteves, M.A.; Damacena, G.N.; Viacava, F. Socio-demographic determinants of self-rated health in Brazil. Cad Saude Publica. 2005, 21 (Suppl), 54-64. Idler, E.L.; Russell, L.B.; Davis, D. Survival, functional limitations, and self-rated health in the NHANES 1 epidemiologic follow-up study, 1992. Am J Epidemiol 2000, 152(9), 874-883.

Discussion: The authors failed to explain how the outcome of the study specifically benefit community/ public health. 

Answer: The information about how the outcome of the study specifically benefit community/ public health has been included in the Discussion Section.

The understanding of those factors is crucial to adjust health promotion, education and care to special needs. The correlates of SRH are important in terms of specific targeted prevention. This is especially important to special populations such as beneficiaries of government welfare assistance, which is ignored or overlooked in most of the public health programs. The results may have implications for public health policy as many of those predictors influencing SRH can be reduced or changed by the specific targeted interventions. The knowledge about the correlates of SHR can help public health professionals priorities health promotion and education. This suggest the need for actions aimed at preventing and diagnosis chronic diseases, promote healthy lifestyle and prevent disabilities. Public health strategies to promote healthy lifestyles or disease prevention interventions can be performed to improve personal health.

Our findings suggest that SRH should be integrated in national health care system and policy making process, and can be used by Polish civil authorities because it will help to predict health problems in population and develop appropriate public health policies and programmes allocate of the resources effectively and identify the areas or populations e.g socially-disadvantaged people and inequity that require special attention to improve the overall health of the population.

Reviewer 2 Report

Overall this is a well written and well designed study, which should be of interest to the readership of the journal.  A notable strength is that this is the first analysis of self-rated health in Poland, and results are largely in line with findings from other countries.  I do have a few minor issues that should be addressed before the paper can be accepted.

Several minor concerns with spelling or grammatical errors in a few places.

In the abstract, line 11, the word "corelatecorrelates" should be changed to "correlates". Page 2, line 56, "The aims of this study was study to analyze..." should be changed to "The aim of this study was to analyze..." Page 3, lines 99-100, "Significance level p<0.05 was used for applied to statistical interference." should be changed to "The significance level of statistical inference was set at p<0.05."  The headers in tables 2 and 3 still contain words in Polish, which should be changed, I think, to Men and Women, respectively.

One major concern is the fact that this study examines only one district within Poland.  I wonder to what extent this district may differ from the rest of Poland, and the extent to which such differences may limit external validity of the findings.  Also, this is a cross-sectional study, so should address potential limitations due to this design as well.  I think both limitations should be addressed in 2-3 sentences, which should be inserted after line 246 in the manuscript.  Something like this: "Our study is not without limitations.  First, the study used a cross-sectional, which provides measures at only one period in time and may not adequately represent measures of self-rated health and other measures which can change over time.  Second, this study was limited to the Piotrkowski District, which may limit the generalizability of the findings to the larger population of Poland."

Author Response

Reviewer II

Overall this is a well written and well designed study, which should be of interest to the readership of the journal.  A notable strength is that this is the first analysis of self-rated health in Poland, and results are largely in line with findings from other countries.  I do have a few minor issues that should be addressed before the paper can be accepted.

Answer: Thank you.

Several minor concerns with spelling or grammatical errors in a few places.

In the abstract, line 11, the word "corelatecorrelates" should be changed to "correlates".

Answer: This has been changed.

Page 2, line 56, "The aims of this study was study to analyze..." should be changed to "The aim of this study was to analyze..."

Answer: This has been changed.

Page 3, lines 99-100, "Significance level p<0.05 was used for applied to statistical interference." should be changed to "The significance level of statistical inference was set at p<0.05."

Answer: This has been changed.

The headers in tables 2 and 3 still contain words in Polish, which should be changed, I think, to Men and Women, respectively.

Answer: This has been changed.

One major concern is the fact that this study examines only one district within Poland.  I wonder to what extent this district may differ from the rest of Poland, and the extent to which such differences may limit external validity of the findings.  Also, this is a cross-sectional study, so should address potential limitations due to this design as well.  I think both limitations should be addressed in 2-3 sentences, which should be inserted after line 246 in the manuscript.  Something like this: "Our study is not without limitations.  First, the study used a cross-sectional, which provides measures at only one period in time and may not adequately represent measures of self-rated health and other measures which can change over time.  Second, this study was limited to the Piotrkowski District, which may limit the generalizability of the findings to the larger population of Poland."

Answer: The information about the limitations of the study have been included in the Discussion Section. One is that cross-sectional surveys tend to make observations at a single point in time, which is the inability to observe the analyzed associations or determinants among the disadvantaged population over their lifetime.

The study was limited to the Piotrkowski District, which may limit the generalizability of the findings. The Piotrkowski District was chosen because according to United Nations Development Programme - UNDP, in 2012 the Piotrkowski District was located in the area of 39 districts in Poland with the lowest value of development index and health index. Piotrkowski District, according to UNDP analysis belongs to the group of 30 poorest districts in Poland. In my opinion the poorest districts in Poland will have similar problems as those in Piotrkowski district. So it may limit the ability to generalize the results to the general population, not for people with the poorest area.

Additionally I have added the strengths of the study in the Discussion Section.

References:

United Nations Development Programme. National Human Development Report, Poland 2012: Regional and local development. Warsaw 2013. [online] http://issuu.com/undp_poland/docs/lhdi_report_poland_2012_eng]

Reviewer 3 Report

I found the paper interesting and new in the Polish culture. However, the language is not always precise and I have some methodological concerns on results. The tables are too long and could be simplified a bit to clarify them more.

Abstract

There is a mistake at the second line: “The present study was carried out to evaluate the corelatecorrelates (did you mean correlates?) of self-11 rated health status among beneficiaries of social care in Poland.

Study population

What about exclusion criteria? What comment could you give for participation rate? Were the interviews have placed in presence on by telephone?

No information was given about the relationship status of the participants

At line 136 tale 3: did you intend men and women in the label?

Results

The analysis adopted chi-square statistics, but some cells in the several categories are really reduced and could this influence your results? What about this possible statistical bias?

Author Response

Reviewer III

I found the paper interesting and new in the Polish culture. However, the language is not always precise and I have some methodological concerns on results. The tables are too long and could be simplified a bit to clarify them more.

 Answer: Thank you.

The English grammar and language has been corrected by the native speaker but also by professional service (http://www.englishprep.pl/korekta.html) which correct the text based on the standards of The Society for Editors and Proofreaders (http://www.sfep.org.uk/).

I have revised the manuscript according to the Reviewer suggestions and comments. All the comments have been carefully addressed. I responded to results problems below.

Regarding the Tables- as the aim of the paper was to examine the correlates of SRH we want to assess many different predictors (health predictors, sociodemographic predictors and lifestyle predictors) that is why the Tables presenting crude and adjusted models are long. But in my opinion this is the advantage of the paper. Most studies assess one or two determinates of SHR whereas in this article we propose the complex approach.

Abstract

 There is a mistake at the second line: “The present study was carried out to evaluate the corelatecorrelates (did you mean correlates?) of self-11 rated health status among beneficiaries of social care in Poland.

 Answer: This has been corrected. Thank you.

Study population

What about exclusion criteria? What comment could you give for participation rate? Were the interviews have placed in presence on by telephone?

 Answer: The study population consists of social care beneficiaries adults aged 18–59 from Piotrkowski District. This District has a low index of development, especially social development and is in 11th place in Poland. Among 11,867 social care beneficiaries in Piotrkowski District 3636 people were in the age category 18–59 and 50% of them agreed to participate in the study (n=1817). The information about health status was available from 1710 participants. Face to face interviews were conducted with response rate 50%.

Only age was the exclusion, we have recruited only adults aged 18–59.

All the requested information has been included.

No information was given about the relationship status of the participants

 Answer: 84% of the study subjects declared cohabitation with partner and/or family and this variable have no effect on self-rated health. The was no information about marital status, but the question about cohabitation with partner and/or family can be used as similar.

At line 136 tale 3: did you intend men and women in the label?

 Answer: This has been corrected. Thank you.

Results

The analysis adopted chi-square statistics, but some cells in the several categories are really reduced and could this influence your results? What about this possible statistical bias?

Answer: The cells categories are the same in the characteristics of the study population and in the statistical models. The variables such as: age (<30, 30-39, 40-49, 50-59), education (Primary, Vocational, Secondary, High), employment status (Permanent job, Temporary job, Disabled or retired, Unemployed), Cohabitation with partner and/or family (Yes, No (living alone)), Alcohol consumption (Yes, No), Binge drinking (Never, Few times per year, Once per month, Once per week, Don’t know), Recreational physical activity (Yes, No), Current smoking (Yes, Yes occasionally, No), Diet (Dietary Quality Score: Healthy dietary habits, Average dietary habits, Unhealthy dietary habits), Subjective assessment of life satisfaction (Extremely satisfied/Satisfied, Neutral, Slightly dissatisfied, Dissatisfied/Extremely dissatisfied), Number of health problems (None, 1-3, 4-6, >7), Heart attack (Yes, No), High blood pressure (Yes, No), Diabetes (Yes, No). Only in case of Subjective assessment of monthly income and Subjective assessment of living one category was missed “Difficult to say”. Now the category “Difficult to say” have been added to the model and is not statistically significant in adjusted models. So it does not biases the results as the category “Difficult to say” was compared with category living conditions ‘fair/rather fair’ and subjective assessment of living conditions ‘sufficient to cover all living needs plus may save a certain amount’. The rest categories of variables are the same.

Round 2

Reviewer 1 Report

The authors have responded adequately to most of my concerns from the previous review. There are still minor concerns for English and grammar.

For example, 

Line 54: Correct 'SHR' to SRH.

Line 65: Correct "The aims of this study was to analyze..." to "The aims of this study were to analyze".

Line 67: Correct "...this is one of the first study..." to "...this is one of the first studies.."

Line 252: Correct "First, the study used a cross-sectional,..." to "First, the study used a cross-sectional design,..."

Check manuscript for more typos and errors in grammar.

Best of luck!

Author Response

The authors have responded adequately to most of my concerns from the previous review. There are still minor concerns for English and grammar.

Answer: Thank you.

The English grammar and language has been corrected by the native speaker but also by professional service (http://www.englishprep.pl/korekta.html) which correct the text based on the standards of The Society for Editors and Proofreaders (http://www.sfep.org.uk/).

For example, 

Line 54: Correct 'SHR' to SRH.

Answer: This has been corrected.

Line 65: Correct "The aims of this study was to analyze..." to "The aims of this study were to analyze".

Answer: This has been corrected.

Line 67: Correct "...this is one of the first study..." to "...this is one of the first studies.."

Answer: This has been corrected.

Line 252: Correct "First, the study used a cross-sectional,..." to "First, the study used a cross-sectional design,..."

Answer: This has been corrected.

Check manuscript for more typos and errors in grammar.

Answer: The manuscript has been checked for typos and errors in grammar.